# Pretreatment [^18^F]FDG PET/CT Prognostic Factors in Patients with Squamous Cell Cervical Carcinoma FIGO IIIC1

**DOI:** 10.3390/diagnostics11040714

**Published:** 2021-04-16

**Authors:** Ewa Burchardt, Wojciech Burchardt, Paulina Cegła, Anna Kubiak, Andrzej Roszak, Witold Cholewiński

**Affiliations:** 1Department of Radiotherapy and Oncological Gynecology, Greater Poland Cancer Center, 61-866 Poznan, Poland; andrzej.roszak@wco.pl; 2University of Medical Science Poznan, 61-866 Poznan, Poland; wojciech.burchardt@wco.pl (W.B.); witold.cholewinski@wco.pl (W.C.); 3Department of Brachytherapy, Greater Poland Cancer Center, 61-866 Poznan, Poland; 4Department of Nuclear Medicine, Greater Poland Cancer Center, 61-866 Poznan, Poland; paulina.cegla@wco.pl; 5Department of Epidemiology, Greater Poland Cancer Center, 61-866 Poznan, Poland; anna.kubiak@wco.pl

**Keywords:** positron emission tomography computed tomography, FIGO IIIC1 cervical cancer, squamous cell carcinoma

## Abstract

Purpose: This study aims to determine whether semiquantitative parameters obtained from both the primary tumor and metastatic pelvic lymph nodes (PLN) diagnosed in fluoro-18-deoxy-glucose positron emission tomography (FDG-PET-CT) are associated with disease-free survival (DFS), local control (LC), distant metastasis-free survival (DMFS) and overall survival (OS) in patients with locally advanced squamous cervical cancer (LACC) and metastatic pelvic lymph nodes. Materials: Retrospective analysis was performed on 93 female patients with FIGO IIIC1. The median age was 53 years (27–75). The PET parameters both in the primary tumor and metastatic pelvic lymph nodes, including SUVmax, SUVmean, TLG, MTV, heterogeneity, along with clinical variables, before radical cisplatin-based radiochemotherapy (RCT) were analyzed. The *p*-values < 0.05 were considered statistically significant. Results: Median follow-up was 38 months (4.5–92.6). Three years and five years OS were 75% and 70% respectively. Patients with SUVmax above 12.6, SUVmean above 7.6 and with TLG in tumors >245.7 lived longer (*p* < 0.05). The higher SUVmax or SUVmean reduced increased DMFS (HR 0.3 95%CI 0.56–0.96 and 0.59 95%CI 0.37–0.93). The clinical factors and other FDG PET CT parameters were not found to be statistically relevant in terms of OS, DFS, DM and LC. Conclusions: This study is the first report showing that in LACC patient population with PLN involvement treated with definitive RCT, high SUVmean, SUVmax and TLG of the primary tumor in FDG-PET-CT were linked with longer OS. Lower SUVmean and SUVmax were linked with shorter DMFS. None of the clinical factors and the nodal FDG-PET-CT parameters influenced the outcome.

## 1. Introduction

According to the World Health Organization (WHO), cervical cancer (CC) continues to be a significant public health problem and the annual number of new cases of CC is expected to increase from 570,000 to 700,000 over the next 12 years (https://www.who.int/reproductivehealth/topics/cancers/en/, accessed on 9 February 2021). Cervical cancer ranks 4th place in mortality and morbidity worldwide [1]. Still, a lot of newly diagnosed cervical cancer patients are in the inoperable stage. Standard of care in locally advanced cervical cancer (LACC) is radical radiochemotherapy (RCT). In 2018 a new classification of Federation of Obstetrics and Gynecology (FIGO) staging introduced lymph node status as a factor of advanced disease (see Appendix A, FIGO Staging 2018). Patients with positive metastatic lymph nodes below aorta bifurcation (pelvic lymph nodes, PLN) are classified as stage IIIC1. As accurate staging is crucial in treatment planning and has an impact on prognosis and positron emission tomography/computed tomography with 2-deoxy-2-[^18^F]fluoro-D-glucose ([^18^F]FDG PET/CT) imaging is recommended before radical treatment (https://www.nccn.org/professionals/physician_gls/pdf/cervical.pdf, accessed on 9 February 2021). Response to RCT remains one of the most critical factors associated with local recurrence and overall survival (OS); however, factors associated with failure are poorly known. A more tailored approach based on pre-treatment prediction of recurrence may allow choosing the personalized treatment for patients with more aggressive CC. Different histopathological subtypes and stages can influence treatment outcome even cross the same stage and metabolic results of the [^18^F]FDG PET/CT examination. This study aims to determine whether semiquantitative parameters obtained from both the primary squamous cell carcinoma (SCC) tumor and metastatic PLN seen in [^18^F]FDG PET/ CT are associated with disease-free survival (DFS), local control (LC), distant metastasis-free survival (DMFS) and OS.

## 2. Materials and Methods

### 2.1. Patients Characteristic

A retrospective analysis was performed on a group with newly diagnosed, histologically proven CC patients admitted to Department of Radiotherapy and Oncological Gynecology between May 2010 and December 2017. One hundred forty-seven of them were found to have positive distant lymph nodes (LN) and/or paraaortic and/ or pelvic lymph nodes in [^18^F]FDG PET/CT and ninety-three of them were enrolled in this single-center study. They met the inclusion criteria: PLN below aorta bifurcation, SCC type of cancer, and locally advanced, inoperable disease with definitive RTC treatment. FIGO staging was based on a gynaecological examination with ultrasound imaging performed by two specialists in gynaecology and oncology and represented as follows: IIA-1, IIB-39, IIIA-2, IIIB-48, IVA-3 patients. After the revision of new FIGO staging criteria, all those patients were reclassified in IIIC1. All patients underwent routine clinical assessment, including the recording of medical history reviews, complete blood count, blood chemistry tests. All patients also underwent [^18^F]FDG PET/CT for initial diagnosis, staging, and radiotherapy (RT) planning. Patients underwent external beam radiotherapy (EBRT) combined with chemotherapy. RT was administered to the whole pelvic region and lymphatic chain up to aorta bifurcation in 28 fractions of 1.8 Gray (Gy) with a total dose of 50.4 Gy EBRT was delivered in a three-dimensional conformal beams method and concomitant chemotherapy with cisplatin 40 mg/m^2^ was administered during RT. The image-guided adaptive intracavitary high dose brachytherapy (BT) with magnetic resonance imaging (MRI) was introduced at the end of treatment. Medical data extracted from computerized medical records included all follow-up data, recurrence, and survival status at the end of the study.

### 2.2. PET Imaging and Analysis

Scans were acquired on Gemini TF PET/CT scanner (Philips Healthcare Medical Systems, Inc, Cleveland, OH, USA), on average 60–75 min after IV injection of [^18^F]FDG with the mean activity of 364 ± 75 MBq. The examination was performed in patients who were fasting for at least five hours before the test (median glucose level: 88 mg% range 56–160 mg%). The PET/CT imaging was done according to our study protocol in the bed position (1.3 min for one bed position) and cutting scans 5 mm thick. The optimal contouring method was obtained using the 35% SUV_max_ (standardized uptake value) threshold and was already verified and described in our previous research [2]. To determine heterogeneity of the primary tumor area under Cumulative SUV-volume histograms the tumor heterogeneity index (AUC-CSH) was used. All PET/CT scans were fused and tumor volumes for cervix and PLN measured using a dedicated workstation in all planes.

The selected ROI (region of interest) was manually adjusted by visual inspection to exclude adjacent FDG-avid structures. Maximum standardized uptake value (SUV_max_), mean standardized uptake value (SUV_mean_), metabolic tumor volume (MTV), and total lesion glycolysis (TLG) of the cervical tumor were recorded during [^18^F]FDG PET/CT scans. The parameters were calculated as follows: SUV = radioactivity of the sensitive area/ratio of the injected dose to the patient’s weight, SUV_max_ was the maximum SUV in the ROI, MTV was the volume included in the curve bigger than or equal to Th35% SUV_max_, SUV_mean_ was the mean SUV in the MTV, and TLG was calculated as SUV_mean_ × MTV. Beside parameters for primary tumor, we also assessed parameters which include the sum of values from primary tumor and metastatic PLN: TLGtotal, SUVtotal and MTVtotal (Figure 1).

### 2.3. Follow-Up

Follow-up included a clinical examination every three months for the first two years and every six months for the following three years or longer. Imaging (MRI, computed tomography CT, PET/CT, vaginal ultrasound) complementary to the clinical examination was performed as part of the follow-up. The last date of follow up was September 2019, but survival data were verified in Regional Cancer Registry and the National Cancer Registry (NCR) on the date December 2018.

### 2.4. Statistical Analysis

All data were gathered in the MS Excel spreadsheet, and descriptive statistics were developed. DFS was defined as the time from the start of treatment to the date of the first clinical or imaging findings that suggested disease recurrence: either distant metastasis or local relapse. Both local recurrence and systemic failure were defined as the time from the start of treatment to the day of local/systemic relapse recognized in an examination and imaging and/or proven by biopsy. OS was defined as the time from the start of treatment to death. The continuous variables cervical and lymph nodes metabolic parameters were dichotomized at the median and the Kaplan-Meier method was used in the survival analysis, and the F cox test was used as a comparison test of two groups. If the F Cox test was inappropriate log rang test was used. The Cox proportional hazard model was used to evaluate prognostic variables. Statistical analyses were performed with Statistica 12 software (StatSoft Inc.), and *p*-values <0.05 were considered statistically significant.

## 3. Results

Ninety-three patients were included in the study. The median age was 53 years at diagnosis (range 27–75 y), the median BMI was 26 (range 12.4–37.9) and the median time of hospitalization was 58 days (range 31–90). The included patients had a biopsy with proven SCC. Median total dose (TD) for the primary tumor was 50 Gy with EBRT (range 45–50.4 Gy) and 28 Gy (range 14–28 Gy) with BT. Median TD for not active in [^18^F]FDG PET imaging PLN was 50.4 Gy (range 45–50.4 Gy). Positive PLN were boosted in 49 cases. Median cisplatin courses were 4 (range 1–7) (Table 1).

Median follow-up was 38 months (range 4.5–92.6). A total of 73% of patients were alive (Figure 2A). Estimated 3-year and 5-year OS were 75% and 70%, respectively. Local recurrence was observed in 14 cases, 5 of them are still alive. Estimated 3-year and 5-year LC was 82.2% and 80.9% respectively, based on examination and imaging (Figure 2B, Figure 3A,B and Figure 4A,B). Distant metastases were observed in six cases; two of them are still alive. A total of 87% developed no systemic failure. Two patients had both local failure and distant metastases diagnosed at the same time. 

Disease-free survival was observed in 68 cases (71%) (Figure 2C). Twenty-five patients died during follow-up and 11 of them were diagnosed and reported in our ambulatory as oncological reason, either local or systemic failure (Figure 2D). The next seven patients were reported in NCR as death due to cervical cancer, but in the ambulatory control visit 2–3 months earlier, the relapse was not diagnosed. These patients are included in DFS analysis but in neither local control analysis nor in distant metastases analysis. Seven patients died due to non-oncological reasons: bowel perforation (*n* = 2), pancreatitis (*n* = 2), peritonitis (*n* = 1) and two patients without a known reason (with good disease control and without any oncological symptoms two months earlier in ambulatory control visit).

All patients showed an increased [^18^F]FDG uptake in the cervical tumors detected at PET/CT (*n* = 93). In this group, semiquantitative [^18^F]FDG parameters for the cervical tumor SUV_max_, SUV_mean_, MTV, and TLG were 12.89 (range 4.56–32.61); 7.59 (range 2.58–17.93); 34.01 mL (range 7.00–115.25) and 245.64 (range 26.0–1667.7), respectively. The mean AUC-CSH index was 0.58 (range 0.52–0.71) and heterogeneity 0.27 (range 0.17–0.32) (Table 2). FDG PET CT parameters of patients free of disease and FDG PET CT parameters of patients with reccurence are shown in Appendix A (see Appendix A).

### 3.1. Overall Survival Analysis

In the analysis of OS, the patients with SUV_max_ above 12.6 (*p* = 0.01; Figure 5A), SUV_mean_ above 7.6 (*p* = 0.01, Figure 5B) and with TLG ≥ 245.7 (*p* = 0.03 Figure 5C) lived longer. Patients with SUV_max_ ≥ 12.6 and SUV_mean_ ≥ 7.6 lived 42 months and with SUV_max_ < 12.6 and SUV_mean_ < 7.6 lived 37.9 months. 

In other PET parameters, no significant differences were noticed compared to OS. Moreover, in the distant metastases free group, all patients with metastasis had SUV_max_ < 12.6 and SUV_mean_ < 7.6 and had shorter time to develop metastasis (Figure 6A,B), which corresponds with OS analysis. No differences were observed in analyzed dichotomized [^18^F]FDG PET parameters between patients with LC or DFS. 

Among the [^18^F]FDG PET/CT pre-treatment parameters in univariate analysis in terms of OS, only SUV_max_ and SUV_mean_ were significant parameters (Hazard Ratio (HR) 0.88, 95% confidence interval (95%CI) 0.8–0.97 and HR 0.81, 95%CI 0.69–0.95 *p* < 0.05, respectively (Table 3)). Higher SUV_max_ and higher SUV_mean_ were linked with longer survival. The increase in the SUV_max_ or SUV_mean_ by one unit was linked with decreased risk of death by 12% and 18% in the whole group, respectively (Table 3). Both variables are correlated together, so they could not be used for multivariate analysis. To show a significant influence on survival in univariate analysis, TLG parameters were dichotomized using a median value of 245.7. Patients with TLG above 245.7 had 60% greater chance of survival in univariate analysis HR 0.4 (95%CI 0.17–0.92, *p* < 0.05). 

### 3.2. Distant Metastasis-Free Survival Analysis

Tumor SUV_max_ and SUV_mean_ were significant parameters in univariate analysis with time to DM. The higher SUV_max_ or SUV_mean_ reduced the risk of DM (HR 0.73 95%CI 0.56–0.96 and 0.59 95%CI 0.37–0.93), which corresponded with the results from Kaplan-Meier survival analysis (Table 3). Other analyzed [^18^F]FDG PET/CT parameters were not different in terms of distant metastases free survival.

Three-year distant metastasis-free survival was 92.3%, and 5-year was 89.8%. There was no significant model in the univariate analysis of [^18^F]FDG PET/CT parameters in terms of DFS or LC analysis. The clinical factors (age, grade, FIGO, time of hospitalization, cycles of chemotherapy) were not found to be statistically relevant for the outcome in terms of OS, DFS, DM and LC (*p* > 0.5, Table 3).

An example of a woman with stage IIIC1 cervical cancer before and full response after radiochemotherapy. 

## 4. Discussion

The parameters SUV_max_, SUV_mean_, TLG obtained from the primary tumor seen in [^18^F]FDG PET/CT influenced OS. Higher SUV_max_, SUV_mean_ of the primary tumor were associated with longer distant metastasis-free survival (DMFS). None of the other PET/CT parameters were found to be prognostic in terms of survival or disease control. 

Issues regarding prognosis of locally advanced adenocarcinoma (AC) and SCC of the cervix have often been raised in the literature. The SCC and AC have many histological and genetics differences what has impact on cell metabolism, treatment response and prognosis. In some reports, the clinical outcome of these two subgroups of patients varied by 10% in favor of SCC [3,4]. Huang also reported a lower five-year relapse-free survival rate for 148 patients with AC [5,6]. In early advanced disease, there is a significant difference of 6% in 10-year DFS and 8% in 10-year OS between SCC and AC pts in favor of SCC [7,8]. Additionally, there was evidence SUV_max_ is higher in squamous cell carcinomas compared to other histopathological diagnosis [9,10]. Therefore, for clarity of analysis, we focused on the very homogenous SCC tumors and parameters in relation to OS, DMFS, DFS and LC (Graph 1) [11].

There is a new clinical FIGO classification for involved PLN, and it is still unclear whether IIIC1 patients should be treated differently. We analyzed patients who were qualified for treatment according to the previous long-valid classification and we did not observe the influence of FIGO stage of primary tumor on OS, DMFS, DFS and even LC (Table 3). Therefore, our data supports the new FIGO classification for this group of patients. In our FIGO IIIC1 group, patients with higher SUV_max_ (≥12.6, Graph 2A) and SUV_mean_ (≥7.6, Graph 2B) lived longer. Moreover, these patients were less likely to have distant metastases, which probably resulted in the fact that we did not observe this relationship in DFS. The negative prognostic value of SUV_max_ was evaluated in the study of Kidd et al., who showed in a predictive model for CC that SUV_max_ was insignificant for OS (HR 1.009 and 95% confidence interval was 0.977–1.042), but significant for recurrence-free survival (RFS) (HR 1.033 and 95% confidence interval was 1.005–1.062). In another work of 93 patients, the significance of primary SUV_max_ on both RFS and OS was confirmed [12]. However, this study presented some limitations, such as varying histopathologic status. Furthermore, only 33 patients presented active PLN in [^18^F]FDG PET/CT. To metastasize to the lymph nodes, the tumor must undergo many genetic changes, which can significantly affect cell metabolism [13]. Perhaps this effect was observed in the [^18^F]FDG PET/CT scans in our patients. It was partly confirmed by works that also did not observe the impact of high SUV_max_ or SUV_mean_ on patient prognosis. Yoo et al. looked for prognostic values in FDG PET/CT among 73 CC patients. Neither SUV_max_ nor SUV_mean_ of primary tumors correlated with disease-free survival or progression-free survival. In his work only TLG ≥7600 with HR 2.981 obtained from nuclear imaging resulted in multivariate analysis as prognostic factor. 

Imaging is the cornerstone in diagnosis, has a vast potential, and could allow for visualization of different biological processes all together within the entire tumor [14]. Several studies have indicated the tumor hypoxia as an adverse factor in cancer treatment and its association with poor outcome in CC. It can be visualized with [^18^F] Fluoromisonidazole PET/CT or MRI [15,16]. Different biological characteristics of the tumor, e.g., blood supply, fibrosis, angiogenesis, coexist with tumor hypoxia and represent the heterogeneity of microenvironment conditions [17,18,19]. They are of interest because they are often associated with aggressiveness in growth or sensitivity to a specific therapy. Another interesting approach is shown in the work of Hillestad et al., where the distribution of different hypoxia levels on dynamic contrast enhanced (DCE) MRI within the CC tumor could distinguish different prognosis [20]. It would be of added value to correlate information from perfusion-weighted MRI and to visualize perfusion of FDG PET CT, which provides information on blood perfusion and glucose metabolism of the tumor. An important complementary study is gene-based classification, which along with imaging data has been shown to predict PFS significantly better in CC [17]. Gene expression signature for assessing hypoxia-related treatment resistance could be used with imaging information to personalize treatment [21]. Moreover, through comprehensive molecular, proteomic, and integrative profiling there is an insight into the biological diversity of CC.

Some authors also highlight the prognostic importance of volumetric parameters. In our study, the median TLG was 245.64 (range 26–1667), and patients with higher TLG unexpectedly lived longer (*p* < 0.05). The effect of TLG on DFS or LC was nonsignificant. Such results are to be expected, because TLG is partly dependent on SUVs. Hong et al. showed that high TLG of the primary lesion is predictive for recurrence-free survival (RFS) in patients with CC. They showed the outcome with only 20 months median follow up of multivariate analysis for recurrence-free survival with TLG ≤215.02 vs. >215.02, HR was 10.172 (95%CU, 1.246–83.044). A negative association on survival with higher MTV revealed a study on 120 pts [22]. All patients included in the study had SCC histopathology, which was a rarity. Unfortunately, the authors did not include nodal status, and treatment types were different. In our homogenous group tumor MTV was not statistically relevant (*p* > 0.05, Table 3) [23].

Lymph node status is known to be a strong independent negative prognostic factor in cervical cancer. The question arises as to whether the LN status is the factor driving the prognosis or it is still the characteristics and the potential for aggressive growth of primary tumor. It seems that promoting the metastatic spread of tumor cells is determined by the nature of the primary lesion. We did not observe any impact of clinical and [^18^F]FDG PET/CT parameters of positive lymph nodes on OS, DMFS, LC DFS (Table 3). Similar results were found in the sub-analysis of 33 patients in the work of Voglimacci et al., which showed the leading role of primary tumor in the group with already defined positive PLN [13].

Our study has some limitations. Despite the retrospective nature and full insight into the history of the disease, there is some missing information in pathology reports like grading or Ki-67 level, HPV status. To reduce the potential error resulting from the analysis of the study in one center, PET images were analyzed by two independent professionals, a specialist in nuclear medicine and RTT experienced in gynecology oncology. 

Concentrating research efforts on identifying specific factors in a noninvasive way before treatment may lead to tailor the right treatment and thus to a reduction in mortality for many cancer patients. Cervical AC is clearly different from SCC based on its molecular pathogenesis, histology, and clinic. Therefore, it should not be analyzed with SCC data, which is underlined by our findings, which show that the high SUV_max_, SUV_mean_ and TLG is linked with longer OS. This can suggest that tumors with high SUV values respond better to radiochemotherapy. Additionally, patients in our group with high metabolic activity have not developed distant metastasis. This phenomenon warrants further study. 

## 5. Conclusions

In FIGO IIIC1, locally advanced cervical cancer patient population and SCC pathology treated with definitive RCT high SUV_mean_, SUV_max_ and TLG of the primary tumor were linked with longer OS. These features should be analyzed in an adaptive approach to treatment. Lower SUV_mean_ and SUV_max_ were linked with shorter DMFS. None of the nodal [^18^F]FDG PET/CT parameters influenced OS, DFS, DMFS and LC. 

## Figures and Tables

**Figure 1 diagnostics-11-00714-f001:**
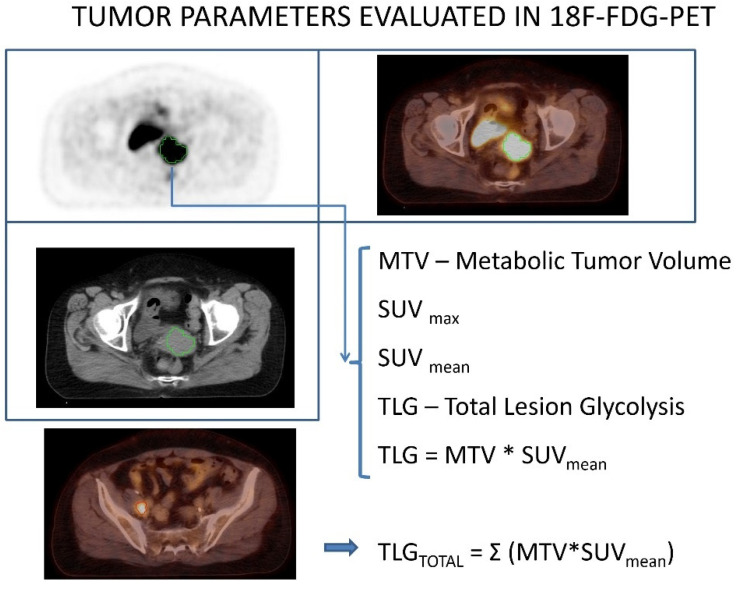
Tumor parameters evaluated in 18F-FDG-PET/CT.

**Figure 2 diagnostics-11-00714-f002:**
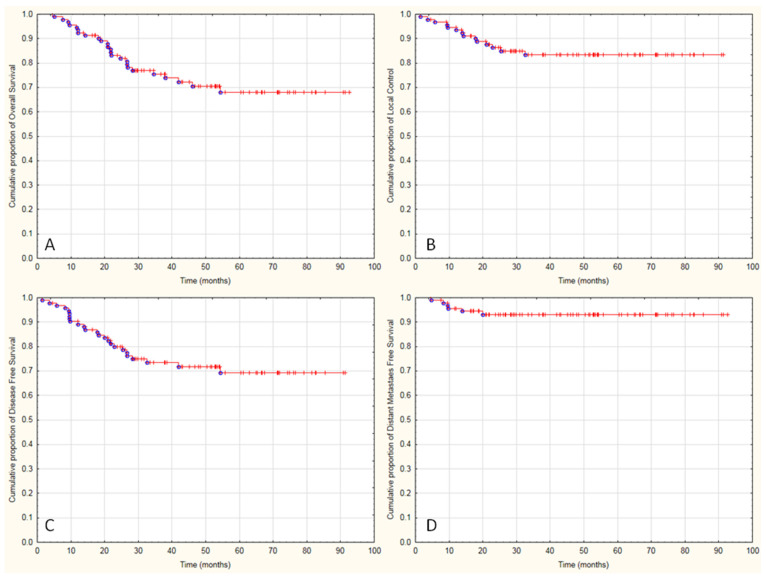
Kaplan-Meier graphs presenting the cumulative proportion of overall survival (**A**), local control (**B**), disease-free survival (**C**), distant metastasis-free survival (**D**) in patients with cervical cancer and positive metastatic pelvic lymph nodes treated with radical radiochemotherapy.

**Figure 3 diagnostics-11-00714-f003:**
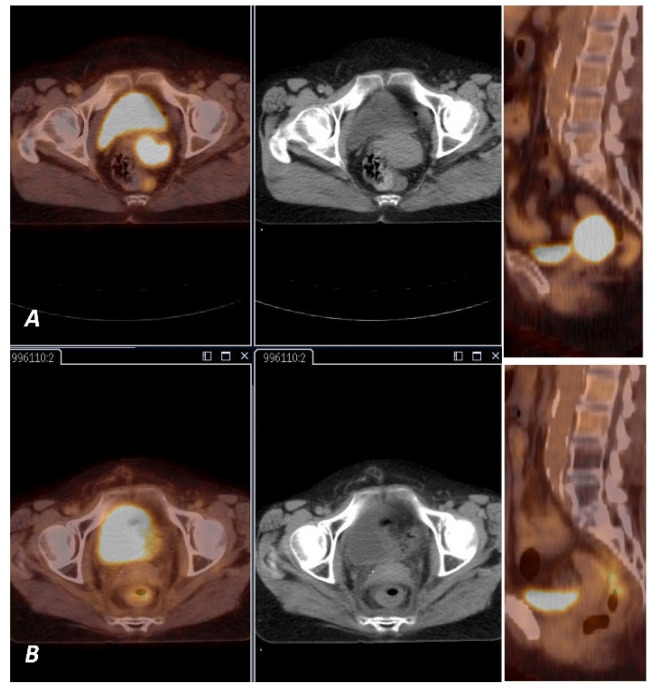
(**A**) ^18^F-FDG PET image shows FDG uptake by tumor in the cervix (arrow). (**B**) ^18^F-FDG PET image shows no FDG uptake in the cervix (arrow).

**Figure 4 diagnostics-11-00714-f004:**
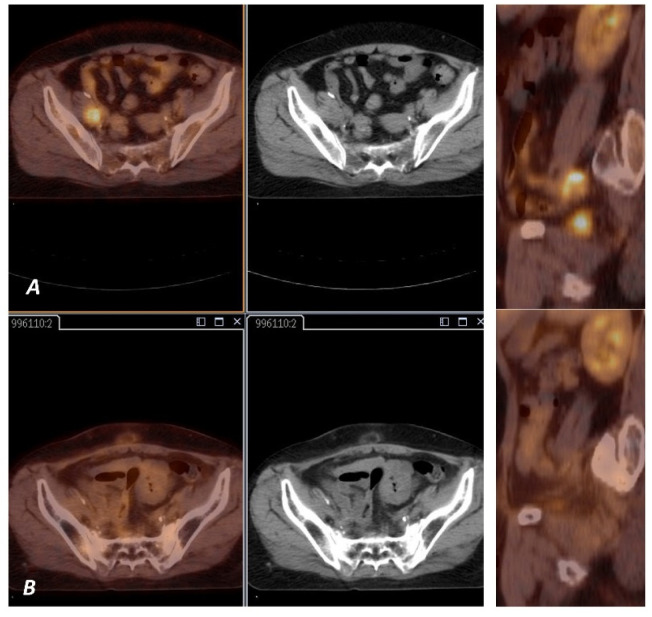
(**A**) ^18^F-FDG PET image shows FDG uptake by the right pelvic lymph node (arrow). (**B**) ^18^F-FDG PET image shows no FDG uptake in the right pelvic lymph node (arrow).

**Figure 5 diagnostics-11-00714-f005:**
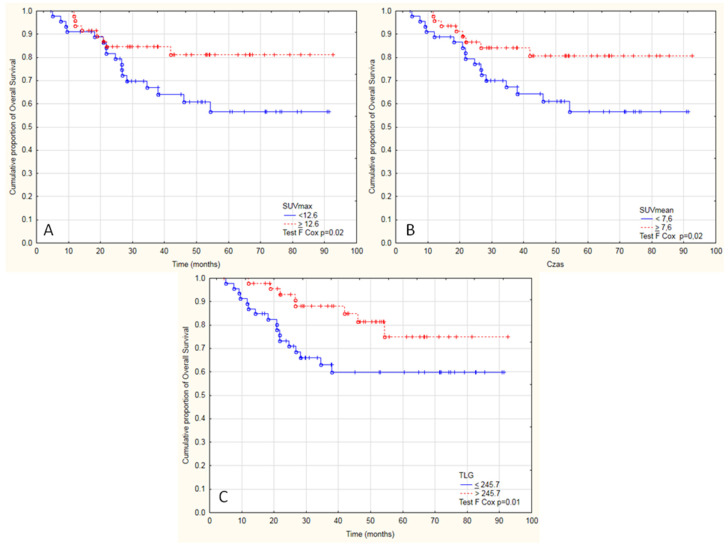
Kaplan-Meier graphs presenting significant differences between the cumulative proportion of overall survival in patients with cervical cancer and positive metastatic pelvic lymph nodes treated with radical radiochemotherapy divide into two groups by (**A**) median SUVmax 12.6 (F Cox test *p* = 0.02), (**B**) SUVmean 7.6 (F Cox test *p* = 0.02), or (**C**) TLG 245.7 (F Cox test *p* = 0.01).

**Figure 6 diagnostics-11-00714-f006:**
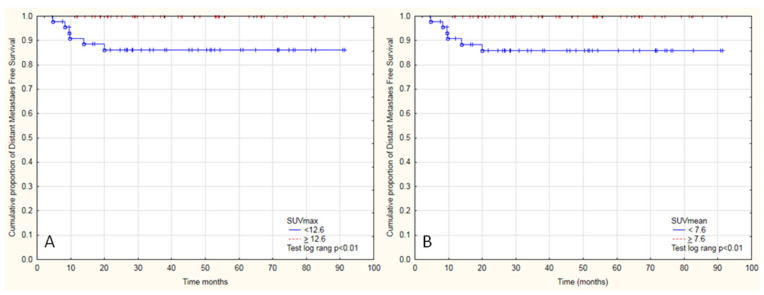
Kaplan-Meier graphs presenting significant differences between the cumulative proportion of distant metastasis-free survival in patients with cervical cancer and positive metastatic pelvic lymph nodes treated with radical radiochemotherapy divided into two groups by (**A**) median SUVmax 12.6 (Test log rank, *p* < 0.01), by (**B**) SUVmean 7.6 (Test log rank, *p* < 0.01).

**Table 1 diagnostics-11-00714-t001:** Patient characteristics.

	*n*	%	Average	Median	Min	Max	SD
Age (Years)	93	100.0	52.4	52.9	27.5	74.8	10.0
BMI	88	94.6	26.2	25.95	12.4	37.9	5.5
FIGO Stage							
IIA	1	1.1					
IIB	39	41.9					
IIIA	2	2.2					
IIIB	48	51.6					
IV	3	3.2					
Length of Hospitalization (Days)	93	100.0	58.1	58	31	90	10.6
SCC Grade							
G1	3						
G2	48						
G3	19						
n.d.	23						
Chemotherapy Courses	93	100.0	4.2	4	1	7	1.2
TD for Elective LN	93	100.0	50.0	50.4	45	50.4	1.0
TD for Primary	93	100.0	47.9	50	45	50.4	2.6
Boost with EBRT for LN+ (Gy)							
Yes	49	52.6	10.1	10	8	19	1.6
No	44	47.4					
Brachytherapy	93	100.0	26.9	28	14	28	3.1
Follow-Up	93	100.0	43.0	38	4.5	92.6	23.6

BMI: Body Mass Index, FIGO: International Federation of Gynecology and Obstetrics Staging System, SCC: squamous cervical carcinoma, TD: total dose, LN: lymph node, EBRT: external beam radiotherapy, Gy: gray, *n*: number, SD: standard deviation.

**Table 2 diagnostics-11-00714-t002:** FGD PET CT parameters.

	*n*	Average	Median	Min	Max	SD
Activity of FDG (mCi)	93	8.945	8.700	4.3000	13.10	1.747
Glucose Level (mg%)	93	90.914	88.000	56.0000	160.00	17.454
SUVmax	93	14.133	12.890	4.5600	32.61	5.510
SUVmean	93	8.199	7.590	2.5800	17.93	3.188
TumorSUV	93	5076.946	3869.440	652.8600	2,6057.90	4230.602
TLG	93	322.353	245.640	26.0300	1667.71	272.469
MTV	93	37.059	34.010	7.0000	115.25	19.791
SUVtotal	93	23.284	21.570	10.6300	45.10	8.845
TLGtotal	93	354.185	258.840	45.5600	1769.20	309.062
MTVtotal	93	40.414	38.750	8.0600	116.56	20.280
SUVLN	93	9.150	7.530	1.9400	26.47	5.716
TLGLN	93	31.833	7.440	0.1000	1667.97	173.599
MTVLN	93	3.356	2.480	0.0500	12.97	2.783
AUC-CSH	93	0.582	0.578	0.5170	0.71	0.037
Heterogenity	93	0.268	0.271	0.1710	0.32	0.025

MTV: metabolic target volume (MTV) of the cervical tumor, MTVLN: MTV obtained from all the metastatic pelvic lymph nodes(PLN), MTVtotal: the sum of MTV of tumor and MTVLN, SD: standard deviation, SUVLN: maximum standardized uptake value of PLN, SUVmax: maximum standardized uptake value of the cervical tumor, SUVmean: mean of standard uptake value of the cervical tumor, SUVtotal: the sum of SUVmax of a tumor and SUVmax obtained from all the metastatic PLN, TLG: total lesion glycolysis of the cervical tumor, TLGLN: total lesion glycolysis obtained from all the metastatic PLN, TLGtotal: the sum of TLG of tumor and TLG obtained from all the metastatic PLN, TumorSUV: the sum of all SUV values.

**Table 3 diagnostics-11-00714-t003:** Univariate analysis of Cox regression.

PET Parameters	Death			Metastasis			Progression			Local Recurrence		
	Hazard Ratio	CI 95%	*p* Value	Hazard Ratio	CI 95%	*p* Value	Hazard Ratio	CI 95%	*p* Value	Hazard Ratio	CI 95%	*p* Value
SUVmax	0.89	0.81–0.98	0.01	0.73	0.56–0.96	0.02	0.98	0.91–1.06	0.57	1.03	0.94–1.13	0.50
SUVmean	0.81	0.69–0.95	0.01	0.59	0.38–0.94	0.02	0.95	0.84–1.09	0.47	1.04	0.89–1.22	0.58
TumorSUV	1.00	1–1	0.42	1.00	1–1	0.09	1.00	1–1	0.96	1.00	1–1	0.81
TLG *	0.40	0.17–0.92	0.03	0.18	0.02–1.57	0.12	0.95	0.43–2.08	0.90	1.10	0.29–4.09	0.89
MTV	1.00	0.99–1.02	0.69	0.97	0.91–1.02	0.22	1.00	0.98–1.02	0.78	1.00	0.98–1.03	0.73
SUVtotal	0.96	0.92–1.01	0.13	0.97	0.88–1.07	0.54	1.00	0.96–1.05	0.97	2.24	0.7–7.14	0.17
TLGtotal *	0.47	0.21–1.07	0.07	-	-	-	1.00	1–1	0.75	1.01	0.96–1.07	0.62
MTVotal	1.00	0.99–1.02	0.65	0.97	0.92–1.02	0.26	1.00	0.98–1.02	0.75	1.00	0.98–1.03	0.71
SUVLN	1.00	0.93–1.07	0.96	1.88	0.34–10.28	0.47	1.02	0.96–1.09	0.54	1.02	0.96–1.09	0.54
TLGLN *	1.02	0.47–2.25	0.95	1.08	0.96–1.22	0.20	1.06	0.48–2.33	0.88	0.71	0.24–2.04	0.52
MTVLN	1.03	0.9–1.18	0.65	1.06	0.82–1.36	0.67	1.02	0.89–1.17	0.75	1.03	0.86–1.23	0.75
Heterogeneity **	1.05	0.89–1.23	0.57	0.96	0.71–1.3	0.78	1.03	0.88–1.2	0.73	0.99	0.81–1.22	0.93
Age	0.99	0.95–1.03	0.58	0.96	0.71–1.3	0.78	1.00	0.96–1.04	0.90	1.00	0.95–1.06	0.88
FIGO Stage	1.37	0.91–2.05	0.13	1.03	0.48–2.24	0.94	1.52	1–2.32	0.05	1.55	0.58–4.1	0.38
Time of Hospitalization	0.97	0.93–1.01	0.19	0.98	0.91–1.06	0.64	1.00	0.96–1.04	0.83	0.97	0.92–1.02	0.24
Tumor Grade	2.19	0.97–4.95	0.06	3.90	0.68–22.24	0.13	1.43	0.59–3.46	0.43	0.70	0.19–2.57	0.59
Boost to PLN+	0.73	0.31–1.7	0.46	1.31	0.27–6.52	0.74	1.48	0.58–3.79	0.41	1.90	0.66–5.51	0.23
Chemotherapy Courses	0.92	0.66–1.27	0.61	1.46	0.7–3.03	0.31	0.99	0.71–1.38	0.96	0.76	0.5–1.15	0.20

CI: confidence interval, FIGO: International Federation of Gynecology and Obstetrics Staging System, HR: hazard ratio, MTV: metabolic target volume (MTV) of the cervical tumor, MTVLN: MTV obtained from all the metastatic pelvic lymph nodes (PLN), MTVtotal: the sum of MTV of tumor and MTVLNSD: standard deviation, SUVLN: maximum standardized uptake value of PLNSUVmax: maximum standardized uptake value of the cervical tumor, SUVmean: mean of standard uptake value of the cervical tumor, SUVtotal the sum of SUVmax of a tumor and SUVmax obtained from all the metastatic PLN, TLG: total lesion glycolysis of the cervical tumor, TLGLN: total lesion glycolysis obtained from all the metastatic PLN, TLGtotal: the sum of TLG of tumor and TLG obtained from all the metastatic PLN, TumorSUV: the sum of all SUV values within the tumor lesion. * the parameter was dichotomized to the analysis. ** the parameter was 100 times increased to the analysis.

## Data Availability

The data presented in this study are available on request from the corresponding author.

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
