# Peer review of "Pretreatment [18F]FDG PET/CT Prognostic Factors in Patients with Squamous Cell Cervical Carcinoma FIGO IIIC1"

_diagnostics, 2021, doi:10.3390/diagnostics11040714_

Round 1
Reviewer 1 Report
Should invite Gyn. Oncologists to join in and listed as co-authors. Because 3-years and 5-years OS data were influenced by their salvage treatment.
FIGO 2018 staging also apply pathological staging. Can you incorporate this into your manuscript?
PET SUV uptake was influenced by patient's renal function which should be addressed.
Author Response
"Please see the attachment."

Reviewer 2 Report
Line 45 : typo on disease
Line 93 : SUV not spelled at first occurrence, AUC/SCH not spelled
IRB approval not specified
Table 1 2 typos on Length of hospitalization
Table 2 might be complemented by the corresponding figures in patients free of disease vs patients who recurrent
Line 189 DFMS not spelled
Line 191 sentence should be rephrased as : “No differences were observed”
Line 198 : important should be replaced by significant
Lines 231-232 duplicate the legend of figure, should be replaced by a call to figures 5 and 6
Line 234 : full, not fully
Line 236 : cervical cervix, please fix
Line 272 : has, not had
Line 281 : study, not studies
Line 282 : not significant instead of insignificant
Lines 285, 287, 288 : “he” refers to Hong et al., hence “they” should be used ; in addition, the argumentation in the whole paragraph is difficult to understand and should be rewritten
Line 291 : Sentence is awkward
Conclusion is a summary of results. Should feature a statement on the clinical utility of the finding : adaptation of treatment ? closer follow up ?
Author Response
The text has been corrected as suggested. "Please see the attachment- Track Changes"."
IRB approval not specified: Decision attached, described below the manuscript
Table 2 might be complemented by the corresponding figures in patients free of disease vs patients who recurrent. Tables attached to manuscript.
